# Nicotinamide Mononucleotide Administration Prevents Experimental Diabetes-Induced Cognitive Impairment and Loss of Hippocampal Neurons

**DOI:** 10.3390/ijms21113756

**Published:** 2020-05-26

**Authors:** Krish Chandrasekaran, Joungil Choi, Muhammed Ikbal Arvas, Mohammad Salimian, Sujal Singh, Su Xu, Rao P Gullapalli, Tibor Kristian, James William Russell

**Affiliations:** 1Department of Neurology, University of Maryland School of Medicine, Baltimore, MD 21201, USA; Kchandrasekaran@som.umaryland.edu (K.C.); jochoi@som.umaryland.edu (J.C.); MArvas@som.umaryland.edu (M.I.A.); mohammad.salimian@gmail.com (M.S.); Sujalsingh33@gmail.com (S.S.); 2Veterans Affairs Medical Center, Baltimore, MD 21201, USA; Tkristian@som.umaryland.edu; 3Department of Diagnostic Radiology and Nuclear Medicine, University of Maryland School of Medicine, Baltimore, MD 21201, USA; SuXu@som.umaryland.edu (S.X.); rgullapalli@som.umaryland.edu (R.P.G.); 4Department of Anesthesiology; University of Maryland School of Medicine, Baltimore, MD 21201, USA; 5Department of Anatomy and Neurobiology, University of Maryland School of Medicine, Baltimore, MD 21201, USA

**Keywords:** NAD^+^, NMN, diabetes, dementia, cognitive impairment, mitochondria, SIRT1, PGC-1α, NEDD4-1

## Abstract

Diabetes predisposes to cognitive decline leading to dementia and is associated with decreased brain NAD^+^ levels. This has triggered an intense interest in boosting nicotinamide adenine dinucleotide (NAD^+^) levels to prevent dementia. We tested if the administration of the precursor of NAD^+^, nicotinamide mononucleotide (NMN), can prevent diabetes-induced memory deficits. Diabetes was induced in Sprague-Dawley rats by the administration of streptozotocin (STZ). After 3 months of diabetes, hippocampal NAD^+^ levels were decreased (*p* = 0.011). In vivo localized high-resolution proton magnetic resonance spectroscopy (MRS) of the hippocampus showed an increase in the levels of glucose (*p* < 0.001), glutamate (*p* < 0.001), gamma aminobutyric acid (*p* = 0.018), *myo*-inositol (*p* = 0.018), and taurine (*p* < 0.001) and decreased levels of *N*-acetyl aspartate (*p* = 0.002) and glutathione (*p* < 0.001). There was a significant decrease in hippocampal CA1 neuronal volume (*p* < 0.001) and neuronal number (*p* < 0.001) in the Diabetic rats. Diabetic rats showed hippocampal related memory deficits. Intraperitoneal NMN (100 mg/kg) was given after induction and confirmation of diabetes and was provided on alternate days for 3 months. NMN increased brain NAD^+^ levels, normalized the levels of glutamate, taurine, N-acetyl aspartate (NAA), and glutathione. NMN-treatment prevented the loss of CA1 neurons and rescued the memory deficits despite having no significant effect on hyperglycemic or lipidemic control. In hippocampal protein extracts from Diabetic rats, SIRT1 and PGC-1α protein levels were decreased, and acetylation of proteins increased. NMN treatment prevented the diabetes-induced decrease in both SIRT1 and PGC-1α and promoted deacetylation of proteins. Our results indicate that NMN increased brain NAD^+^, activated the SIRT1 pathway, preserved mitochondrial oxidative phosphorylation (OXPHOS) function, prevented neuronal loss, and preserved cognition in Diabetic rats.

## 1. Introduction

### 1.1. Diabetes and Neurodegeneration

There is increasing evidence that diabetes predisposes to cognitive decline leading to dementia in both animal models and humans [1]. It is also suggested that Alzheimer’s disease (AD) may represent a consequence of a distinct form of brain specific insulin resistance and impaired glucose regulation. In the Rotterdam study of 6,370 elderly subjects, 126 developed dementia, of which 89 specifically had AD. Type 2-diabetes (T2DM) doubled the risk of a patient having dementia, and patients on insulin had four times the risk [2]. In Diabetic rats, impaired glucose metabolism in the hippocampus tends to shift from the aerobic oxidative metabolic pathway to other pathways [3,4,5,6]. However, the consequences of this change and its connection to diabetes-related cognitive dysfunction remain unknown [7,8,9,10]. Diabetes adversely affects the central nervous system, which involves both acute and chronic metabolic disturbances that can lead to impairment of the functional and structural integrity of the brain [3,6,10,11]. Moreover, oxidative stress plays a central role in brain and neuronal damage in both clinical and experimental diabetes [7,11]. In hyperglycemic conditions, glucose toxicity in neurons [11,12] is mainly due to increased intracellular glucose oxidation, which leads to an increase in reactive species production [7,8]. Further, oxidative damage to various brain regions contributes to development of long-term complications, morphological abnormalities, and memory impairment [13,14,15]. These oxidant radicals contribute to increased neuronal death through oxidation of proteins, DNA damage, and peroxidation of lipids [15].

### 1.2. Diabetes and NAD^+^

Nicotinamide adenine dinucleotide (NAD^+^) is an essential cofactor for multiple cellular redox processes linked to the mitochondrial oxidative phosphorylation system [16,17]. NAD^+^ levels decline with age, in neurodegenerative conditions, after acute brain injury, and in obesity or in diabetes [18,19]. Loss of NAD^+^ results in impaired mitochondrial and cellular functions. NAD(H) furnishes reducing equivalents to the mitochondrial electron transport chain to generate ATP. However, NAD^+^ also acts as a degradation substrate for enzymes such as sirtuins, poly (ADP-ribosyl) transferase 1 (PARP1), and cluster of differentiation 38 (CD38) [20,21,22]. Activation of proteins that deplete NAD^+^ levels (PARP1 and CD38) promote neurodegeneration, while inhibitors of PARP1 and CD38 promote neuronal survival (reviewed in [23]). On the other hand, activators of SIRT1 promote neuronal protection. Knockout of proteins that inhibit SIRT1, such as deleted in bladder cancer protein 1 (DBC1)], promote neuronal survival and axonal regeneration [20,21,22]. Since NAD^+^ is required for the activity of sirtuins, therapeutic interventions with key NAD^+^ intermediates, such as NMN and nicotinamide riboside (NR), which can replenish cellular NAD^+^ levels, have drawn significant attention in the field of aging, obesity, and metabolic syndrome [18,24,25].

### 1.3. Diabetes, SIRT1, and NAD^+^

The K_m_ for NAD^+^ of some sirtuins is high (SIRT1 ~100 µM); these enzymes can act as NAD^+^ sensors (cellular NAD^+^ is 200–500 µM) when there are physiological and pathological changes in NAD^+^ levels, translating metabolic cues into enzymatic and transcriptional adaptations [26]. SIRT1 is a deacetylase enzyme that acetylates transcription factors; prominent amongst these transcription factors are PGC-1α [27,28,29], FoxO family members [30], NF-κB [31], and p53 [32]. Deacetylation of these factors regulates cell death, survival, and energy metabolism. Sirtuins cleave NAD^+^ into nicotinamide and 10-O-acetyl-ADP-ribose [33] or 20- and 30-O-acetyl-ADP-ribose [34] and thereby deacetylate lysine residues. SIRT1 activity requires NAD^+^ [35]. In contrast to other NAD^+^ metabolizing enzymes, activation of SIRT1 by resveratrol protected mice against High Fat Diet (HFD)-induced obesity and insulin resistance [25,36,37].

Use of key NAD^+^ precursors, such as nicotinamide mononucleotide (NMN) and nicotinamide riboside (NR) that replenish cellular NAD^+^ levels can provide therapeutic interventions against diabetes-induced neurodegeneration. A direct effect of NMN administration on diabetes-induced central nervous system neurodegeneration has not been tested. In this study, we tested whether administration of NMN, a precursor to NAD^+^, could prevent diabetes-induced hippocampal impairments. We explored the potential mechanisms of action including regulation of SIRT1 and acetylation of proteins. Our results showed that NMN administration increased brain NAD^+^ levels, prevented loss of SIRT1, and decreased acetylation of hippocampal proteins. These effects prevented the loss of hippocampal function. We propose that NMN administration improves mitochondrial metabolism and promotes protein quality control to support hippocampal memory.

## 2. Results

### 2.1. NMN Normalized Diabetes-Induced Decrease in Brain NAD^+^ Levels

Diabetes was induced in 3-month-old Sprague–Dawley (SD) rats by the administration of streptozotocin (STZ). A week after STZ administration, blood glucose levels were measured to make sure that the levels were above 300 mg/dL, then, nicotinamide mononucleotide (NMN) was administered subcutaneously at a dose of 100 mg/kg on alternate days for the next 3 months. There were 4 groups (*n* = 6 for each group, total = 24): (1) Non-Diabetic; (2) Diabetic; (3) Non-Diabetic + NMN, and (4) Diabetic + NMN. The body weight, blood glucose levels, lipid levels, and brain NAD^+^ levels were measured (Table 1). The baseline measurements in the Non-Diabetic and Non-Diabetic + NMN rats showed no significant differences (Group 1 vs. Group 3 in Table 1). Diabetes increased the plasma glucose level and decreased the body weight, plasma triglycerides, and brain NAD^+^ levels, compared to the Non-Diabetes group (Group 1 versus Group 2 in Table 1). Administration of NMN (100 mg/day) on alternate days to Diabetic rats did not alter body weight or blood chemistry but increased and normalized the diabetes-induced decrease in brain NAD^+^ level (Group 2 versus Group 4 in Table 1). The intraperitoneal glucose tolerance test (*n* = 6) showed a significant increase in area under the curve (AUC) in Diabetic rats compared to Non-Diabetic rats and there was no significant difference in AUC between Diabetic and Diabetic + NMN rats (Table 1 and Appendix A), suggesting NMN did not affect glucose toxicity.

### 2.2. NMN Normalized Diabetes-Induced Changes in Brain Biochemicals

To investigate whether diabetes altered brain biochemicals, we performed in vivo localized high-resolution proton magnetic resonance spectroscopy (MRS) studies on the hippocampus of the brains of the four groups of rats. Magnetic resonance imaging (MRI) revealed no obvious anatomical differences between the four groups. A difference can be observed in the magnetic resonance (MR) spectra of Non-Diabetic and Diabetic groups (Figure 1). The total creatine (creatine + phosphocreatine) tends to be maintained at a relatively constant level and was used as a convenient internal standard (to calculate the ratios for other biochemicals). Three months of diabetes resulted in a significant increase in the hippocampus in the levels of glucose (Non-Diabetic vs. Diabetic = 0.61 vs. 1.07 (*p* < 0.001)), gamma aminobutyric acid (GABA) (Non-Diabetic vs. Diabetic = 0.29 vs. 0.33 (*p* = 0.018)), glutamate (Non-Diabetic vs. Diabetic = 1.16 vs. 1.31 (*p* < 0.001)), myoinositol (Non-Diabetic vs. Diabetic = 1.00 vs. 1. 15 (*p* = 0.018)), and taurine (Non-Diabetic vs. Diabetic = 0.81 vs. 0.91 (*p* < 0.001)) and a significant decrease in N-acetyl aspartate (NAA) (Non-Diabetic vs. Diabetic = 1.22 vs. 1.03 (*p* = 0.002)) and GSH (Non-Diabetic vs. Diabetic = 0.25 vs. 0.17 (*p* < 0.001)). Previous studies showed the same changes in biochemicals in the brains of Type 1 and Type 2 models of diabetes [38,39,40]. The metabolic profile showed that administration of NMN to Diabetic rats decreased glutamate, myoinositol, taurine, and NAA, increased GSH, and normalized the diabetes-induced changes in the biochemicals, except that NMN did not affect hippocampal glucose levels, which remained at the diabetic level (Diabetic vs. Diabetic + NMN = 1.07 vs. 0.97; not significant (NS)); Table 2.

### 2.3. NMN Administration Prevents Diabetes-Induced Hippocampal Changes

#### 2.3.1. NMN Administration Prevents Loss of Hippocampal CA1 Volume

We tested whether hyperglycemia and associated metabolite changes in the brain affect hippocampal structural and neuronal viability. We measured the hippocampal volume and neuronal number in CA1 and dentate gyrus (DG) regions of the hippocampus. There was no significant difference in the total volume of the hippocampus between control rats (215 ± 25 per mm^3^) and Diabetic rats (185 ± 21 per mm^3^, Figure 2d–f). The most significant (40%) decrease in the volume was observed in the CA1 region of the hippocampus in the Diabetic rats (20 ± 3 per mm^3^) compared to Non-Diabetic rats (14 ± 2 per mm^3^; *p* < 0.05). There was no significant decrease in the DG volume between Diabetic rats (54 ± 8 per mm^3^) and Non-Diabetic rats (48 ± 5 per mm^3^). In NMN-treated Diabetic rats, the hippocampal CA1 volume increased significantly from 14 ± 2 per mm^3^ to 19 ± 2 per mm^3^ (*p* < 0.05) and normalized the diabetes-induced decrease. A smaller increase (17%) in CA1 volume was also seen in NMN-treated Non-Diabetic rats compared to that of Non-Diabetic rats (from 20 ± 3 per mm^3^ to 24 ± 2 per mm^3^).

#### 2.3.2. NMN Administration Prevents a Loss of Hippocampal CA1 Neurons

The total number of the neurons in the hippocampus of Non-Diabetic and Diabetic rats was 4.5 ± 0.45 × 10^5^ per µm^3^ and 3.9 ± 0.5 × 10^5^ per µm^3^, respectively; these values were not significantly different from each other (Figure 2G–I). These numbers were obtained by using unbiased stereology and counting 500,000 neurons from Non-Diabetic and Diabetic rat brain sections [41,42]. The mean coefficient of error (CE) of Non-Diabetic and Diabetic rats was 10% and 18%, respectively. On the other hand, there was a significant decrease in the number of neurons in the CA1 region of the hippocampus in Diabetic (1.1 ± 0.1 × 10^5^ per µm^3^) compared to Non-Diabetic rats (1.5 ± 0.15 × 10^5^ per µm^3^; *p* < 0.05). The total number of neurons in the dentate gyrus of the hippocampus in Non-Diabetic rats (3.5 ± 0.5 × 10^5^ per µm^3^) and Diabetic rats (2.5 ± 0.5 × 10^5^ per µm^3^) was not significantly different. In hippocampal CA1, NMN (1.6 ± 0.3 per mm^3^) prevented a diabetes-induced decrease in neurons (1.1 ± 0.1 × 10^5^, *p* < 0.05). A smaller non-significant increase (12%) in CA1 volume was also seen in NMN-treated Non-Diabetic rats (from 1.5 ± 0.2 per mm^3^ to 1.7 ± 0.2 per mm^3^).

### 2.4. NMN Administration Prevents Diabetes-Induced Memory Impairment

Using a Y maze, we determined differences in memory between the four groups of rats. The Y maze assesses rapidly acquired short-term spatial memory and relies on the fact that normal rats prefer novel over familiar spatial environments. For the spontaneous spatial novelty preference test, rats could explore two arms of the maze during the exposure phase, then, a third arm was introduced to see the preference for the novel arm. Discrimination ratio analysis showed that the Diabetic rats had a decreased preference for the novel arm (*p* < 0.05) than did the Non-Diabetic rats. In contrast, the Diabetic + NMN rats had an increased preference for the novel arm compared to that of the Diabetic rats (Figure 3c). In the spontaneous alternation test, the rat could freely explore the three arms (A, B, and C) for 15 min. The number of arm entries and the number of triads were recorded to calculate the percentage of alternation. In the spontaneous alternation test, the number of actual alternations and the number of maximum alternations in the Diabetic rats were significantly lower than those of the Non-Diabetic rats (Figure 3D). The Diabetic + NMN rats showed an improved alternation score percentage.

### 2.5. NMN Administration Increases Mitochondrial Spare Reserve Capacity

To study how administration of NMN, a precursor of NAD^+^, regulates mitochondrial NADH oxidation, namely, oxidative phosphorylation, we measured mitochondrial respiration in hippocampal mitochondria isolated from rat brains of the four groups. Impairment of mitochondrial respiratory chain complex activity has been well documented in diabetic brains [10,18,19]. Measurement of State 4 basal mitochondrial respiration using complex I substrates (malate and glutamate) showed no significant differences in basal respiration between the four groups (Figure 4, Table 3). However, ADP-stimulated State 3 respiration showed a significant increase in hippocampal mitochondria prepared from both Non-Diabetic and Diabetic rats treated with NMN (Table 3; Group1 vs. Group 3 = 109.3 ± 5.6 vs. 119.8 ± 7.5 n mol O_2_/min/mg protein; *p* < 0.05); (Group 2 vs. Group 4 = 82.2 ± 11.8 vs. 103.3 ± 7.5; *p* < 0.01). Similarly, NMN-treatment increased the uncoupled maximal respiration in mitochondria in both Non-Diabetic and Diabetic rats (Table 3 Group1 vs. Group 3 = 272.2 ± 11.3 vs. 302.7 ± 10.3; *p* < 0.05); (Group 2 vs. Group 4 = 240.7 ± 13.6 vs. 295 ± 16.8; *p* < 0.01). Calculation of the spare reserve capacity (uncoupled–basal respiration) was higher (*p* < 0.01) in mitochondria from Non-Diabetic + NMN-treated rats (235 ± 11 n mol O_2_/min/mg protein vs. 204 ± 15; *p* < 0.05) and in mitochondria from the Diabetic + NMN rats (242 ± 17 n mol O_2_/min/mg protein vs. 162 ± 16; *p* < 0.01). These results suggest that diabetes decreases State 3 and uncoupled maximal respiration and that NMN administration primed and protected the mitochondria for high energy demand.

Oxygen consumption rate (OCR) was measured at basal level in the presence of Complex I substrates (malate/glutamate) with the subsequent and sequential addition of ADP, oligomycin, FCCP, and rotenone + antimycin A to hippocampal mitochondria. Basal, oligomycin sensitive, and uncoupled OCR were measured as described in Section 4. Spare respiratory capacity was calculated after subtracting the non-mitochondrial respiration. The significance among the groups is indicated.

### 2.6. NMN Administration Upregulates the NAD^+^-Dependent SIRT1 Signaling Pathway and Downregulates the Acetylation Pathway in the Hippocampus of Diabetic Rats

We determined whether the protein levels of NAD^+^ dependent protein SIRT1 and its targets were altered in the hippocampal protein extracts from the four groups. The NAD^+^-dependent enzymes that we measured were SIRT1, PGC-1α, and acetylated proteins. SIRT1 is a major regulator of mitochondrial biogenesis and metabolism; it activates PGC-1α and in turn promotes mitochondrial biogenesis [27,28,43]. The results showed that there was a significant decrease in both SIRT1 and PGC1-α and an increase in acetylation of proteins in the hippocampal extracts from the Diabetic rats compared to those from the Non-Diabetic rats (Figure 5). Administration of NMN to the Diabetic rats normalized the diabetes-induced decreases in both SIRT1 and PGC-1α and decreased acetylation of neuronal proteins. One of the major de-acetylated protein bands (~102 kDa) was cut and subjected to mass spectrometry analysis. The results showed that the protein that was maximally acetylated was E3 ubiquitin protein ligase NEDD4-1, P46935, molecular weight of 102.71 kDa, and acetylated at positions 124 and 125. NMN treatment deacetylated this protein.

## 3. Discussion

### 3.1. Diabetes and NMN

The findings in this study indicate that subcutaneous administration of NMN prevented the diabetes-induced decrease in NAD^+^ levels, normalized SIRT1 protein levels, and improved downstream activities such as mitochondrial respiration and deacetylation of proteins, which are mediated by NAD^+^, to protect CNS neurons from diabetes-induced neurodegeneration. We propose that improved mitochondrial respiratory capacity via PGC-1α and associated mitophagy via NEDD4 act synergistically to provide neuroprotection. Sirtuins are central to this process. SIRT1 regulates mitochondrial function in hippocampal neurons through PGC1-alpha and activators or overexpression of SIRT1 has been shown to regulate axonal growth [44,45,46]. In addition, NMN also deacetylates mitochondrial proteins via the mitochondrial isoform SIRT3 [47]. In combination, the combined effect of NMN administration is to prevent impairment of memory and hippocampal neurodegeneration.

### 3.2. The NAD^+^ Precursor NMN as a Therapeutic Agent

Therapeutic interventions with key NAD^+^ intermediates, such as NMN and nicotinamide riboside (NR), which can replenish cellular NAD^+^ levels, have drawn significant attention in the field of aging, diabetes, obesity, and metabolic syndrome [16,18,24,25]. In various animal models, the administration of NMN has been shown to alleviate age-associated physiological declines in the liver, adipose tissue, muscle, pancreas, kidney, retina, and central nervous system [48,49,50]. NMN is synthesized from nicotinamide, a form of water-soluble vitamin B3, and 50-phosphoribosyl-1-pyrophosphate (PRPP), by NAMPT, the rate-limiting NAD^+^ biosynthetic enzyme in mammals. NMN is also synthesized from NR via a nicotinamide riboside kinase (NRK)-mediated phosphorylation reaction [18,49,51]. Conversion of NMN into NAD^+^ is catalyzed by nicotinamide mononucleotide adenyl transferases (NMNATs). Systemic NMN administration has been shown to effectively increase NAD^+^ levels in various peripheral tissues, including the brain, suggesting that it crosses the blood brain barrier (BBB). Intraperitoneal NMN administration has been shown to increases NAD^+^ levels in brain regions such as the hippocampus [51,52,53]. Long-term (1 year) oral administration of NMN (up to 300 mg/kg) is not associated with any obvious adverse effects in normal wildtype C57BL/6 mice [48], and recent clinical studies show that NMN is well tolerated in humans [54].

### 3.3. Diabetes-Induced Changes in Biochemicals is Prevented by NMN Administration

Diabetes was induced in rats by the administration of STZ. Three-months of diabetes caused a significant increase in the plasma glucose but a decrease in body weight, plasma triglycerides, and brain NAD^+^ levels compared to the Non-Diabetic rats. Intraperitoneal administration of NMN (100 mg/day) on alternate days to Diabetic rats normalized a diabetes-induced decrease in brain NAD^+^ levels in rats (Group 2 versus Group 4 in Table 1) but did not alter body weight and did not prevent the diabetes-induced increase in plasma glucose. The intraperitoneal glucose tolerance test confirmed that there was no significant difference in AUC between the Diabetic and Diabetic + NMN rats (Table 1 and Appendix A), suggesting NMN did not affect hyperglycemic injury.

Monitoring changes in brain biochemicals provided non-invasive biomarkers for the effect of diabetes and neurodegeneration and elucidated the effects of NMN treatment. The total creatine was used as a convenient internal standard (to calculate ratios for other biochemicals). Measurement of the effects of diabetes and administration of NMN on the metabolic profile showed that levels of hippocampal biochemicals GABA/total Cr, glutamate/total Cr, glutamate + glutamine/total Cr, NAA/total Cr, taurine/total Cr, and mI/total Cr were significantly altered in the Diabetic rats compared to those of the Non-Diabetic rats, suggesting that energy metabolism, neurotransmission, and lipid membrane metabolism were impaired in the Diabetic rats [12]. The brains of Diabetic rats exhibited pathological metabolic changes similar to those observed in Alzheimer’s disease [55]. In the case of GABA, GABAergic inhibition is involved in the control of behaviors, including depression and cognition [56]. One possible explanation of how GABA affects cognition is by means of its inhibitory function on dopamine release [56]. Thus, increased GABA could downregulate the dopamine activity, which might eventually lead to impaired cognitive performance. Administration of NMN normalized the increased levels of GABA and thus could protect against diabetes-induced impairment in hippocampal memory. Glutamate is one of the most important excitatory neurotransmitters and studies have shown that increased glutamate levels are roughly linearly related with increased excitatory activity [57]. Hippocampal CA1 neurons are particularly vulnerable to glutamate toxicity [58]. Thus, increased glutamate could contribute to the loss of hippocampal CA1 neurons in Diabetic rats. Accordingly, our unbiased stereological analysis and behavioral measurements showed a significant cell loss in the CA1 region and cognitive impairment in the Diabetic rats compared to the Non-Diabetic rats. Administration of NMN to Diabetic rats normalized the diabetes-induced increase in glutamate, likely to reduce excitatory activity and protected against CA1 neuronal loss and preserved memory function. N acetyl aspartate (NAA) constitutes ~3–4% of total brain osmolarity and may serve osmoregulatory roles [59]. The reduction in NAA is likely to contribute to an increase in the osmolytes such as myoinositol and taurine. NAA and NAAG are synthesized primarily in the mitochondria, transported to oligodendrocytes, and used for lipid synthesis and myelin generation [60]. A decrease in NAA is an important marker of neuronal loss, mitochondria dysfunction, and demyelination [61].

The current study provides novel evidence that NMN administration prevents diabetes-induced hippocampal neurodegeneration. This observation is supported by the preservation of measures of neurodegeneration, including hippocampal biochemicals, hippocampal CA1 volume, hippocampal CA1 neuronal counts, memory, and hippocampal mitochondrial respiration. (Section 2.1, Section 2.2, Section 2.3, Section 2.4, Section 2.5 and Section 2.6).

### 3.4. NMN Prevents Diabetes-Induced Decreases in Hippocampal Volume and Neuronal Counts

In this study, we found that there was no significant difference in the total volume and neuronal number of the hippocampus between the Non-Diabetic rats and Diabetic rats. Similarly, there was no significant difference in the volume and neuronal number in the dentate gyrus between the diabetic rats and Non-Diabetic rats. Whereas, the CA1 volume and neuronal number in the Diabetic rats significantly declined compared to those of the Non-Diabetic rats. In comparison to dentate gyrus neurons, hippocampal CA1 neurons are more vulnerable to ischemic injury, glutamate excitotoxicity, and type 1 diabetes (reviewed in [58]). It has been suggested that hippocampal CA1 neurons may be intrinsically more vulnerable to mechanical insult than are cortical neurons due to increased densities and lower energy capacities as evidenced by increased susceptibility to mitochondrial dysfunction due to ischemic insults. Furthermore, adult dentate gyrus neurons are more resistant due to increased neurogenesis [58,62,63].

The CA1 region of the hippocampus is involved in cognitive processes, particularly learning, and memory. Spatial memory of rats is strongly dependent on hippocampal activity [64,65]. Loss of hippocampal neurons and damage to the plasticity of synapses plays a key role in the cognitive deficits in central nervous system diseases. In the present study, the novel arm recognition and discrimination ratio in the Diabetic rats was reduced compared to the Non-Diabetic rats, which indicates that STZ-induced diabetes had a negative effect on spatial learning ability and memory. These results agree with previous studies [65,66].

### 3.5. Diabetes-Induced Decreases in Mitochondrial Respiration was Rescued by NMN Administration

In diabetic animals, peripheral blood glucose concentration rises, and peripheral insulin resistance occurs. It is noteworthy that, at the same time, glucose concentration in the hippocampus also rises. This high-glucose environment may inhibit mitochondrial function, leading to massive production of reactive oxygen species [2]. Assessment of mitochondrial bioenergetics showed that both ADP-stimulated oxygen consumption rate and uncoupler-mediated maximal oxygen consumption rate were decreased in hippocampal mitochondria from the Diabetic compared to the Non-Diabetic rats. The decrease in the maximal oxygen consumption rate and significantly lowered spare respiratory capacity suggests that hippocampal neuronal mitochondria are energetically stressed, and that mitochondrial workload is increased in diabetic neuronal mitochondria. On the other hand, administration of NMN increased both maximal oxygen consumption and spare reserve capacity. Spare respiratory capacity represents the amount of extra ATP that can be produced by oxidative phosphorylation in case of a sudden increase in energy demand. Administration of NMN increased spare reserve capacity in hippocampal neuronal mitochondria in both the Non-Diabetic and Diabetic rats. We interpret these results to suggest that administration of NMN primes the mitochondria by increasing their reserve capacity to combat mitochondrial stress under both Non-Diabetic and Diabetic conditions. Regulation of oxidative phosphorylation can be divided into short- and long-term. Short-term regulators mediate the oxidative phosphorylation according to the immediate energy requirement [67]. Short-term regulators are, therefore, important for the regulation of the ATP production. Long-term regulators of oxidative phosphorylation modulate the properties of the oxidative phosphorylation and are correspondingly essential for setting of the maximal respiratory capacity. The increase in State 3, uncoupled respiration, and spare reserve respiratory capacity by NMN treatment is likely to reflect the interplay between the short- and long-term regulators of oxidative phosphorylation. Similar results of increased State 3 (~15%) and uncoupled respiration (~20%) was obtained after NMN treatment (100 mg/kg) in the brains of an AD mouse model [52].

### 3.6. NMN Administration Increases SIRT1 and Deacetylation of Proteins

Our studies suggest that NMN administration enhances the SIRT1/PGC-1α axis in neurons and protects against neurodegeneration in diabetic mice. In addition, NMN activates SIRT3 due to increased intramitochondrial NAD^+^, the mitochondrial proteins are more deacetylated, which leads to increased activity of TCA cycle enzymes, pyruvate dehydrogenase, and increased mitochondrial reserve capacity [51,52,68]. There were higher maximal and spare reserve capacities in hippocampal mitochondria in both the Non-Diabetic + NMN and Diabetic + NMN rats. This is consistent with the concept that sirtuins prime neuronal mitochondria to overcome failure of bioenergetic function induced by an increased glucose load [51,52,68]. SIRT1 is central to regulation of both mitochondrial function and neuronal preservation. For example, SIRT1 uses NAD^+^ to deacetylate proteins, and administration of NMN prevents depletion of cellular NAD^+^ and maintains cellular homeostasis. This occurs by regulating the deacetylation of crucial proteins to maintain quality control via mitochondrial metabolism and by the ubiquitin pathway. In the hippocampal extracts from Diabetic + NMN mice compared to Diabetic mice, the major deacetylated protein was determined to be E3 protein ligase NEDD4-1 (Figure 5c). NEDD4-1 is the most abundant E3 ubiquitin ligase in mammalian neurons. An electron microscopic study observed that knockdown of NEDD4 caused dramatic enlargement of mitochondria, suggesting that NEDD4 may play a pivotal role in mitophagy [69]. We speculate that SIRT1-mediated deacetylation of PGC-α to increase its activity, promote mitochondrial oxidative metabolism, and SIRT1-mediated deacetylation of the E3 protein ligase NEDD4-1 to maintain proteostasis. NMN administration enhances this pathway to protect against diabetes-induced neurodegeneration [69,70,71]. In addition, in mice that express enhanced yellow fluorescent protein (eYFP) in neuronal mitochondria, administration of NMN had far less fragmented mitochondria, but had longer neuronal mitochondria in the CA1 region, suggesting that NMN treatment gives rise to longer mitochondria in the hippocampal sub-region, likely through increased fusion and/or decreased fission [52].

In summary, our results showed that intraperitoneal administration of NMN increased the levels of NAD^+^ in both Non-Diabetic and Diabetic brains. Diabetes, on the other hand, decreased brain NAD^+^ levels and administration of NMN compensated for the loss of brain NAD^+^ and thereby prevented diabetic cognitive impairment and diabetes-induced loss of hippocampal neurons that paralleled preservation of mitochondrial bioenergetic function and activation of the SIRT1 pathway. The study suggests that medications that prevent NAD^+^ depletion induced by diabetes mellitus and molecules that activate SIRT1 may provide a therapeutic intervention for diabetic neurodegeneration in the central nervous system.

## 4. Materials and Methods

### 4.1. Animal Procedure

All animal protocols followed the National Institutes of Health (NIH) Guide for the Care and Use of Laboratory Animals and were approved by the Institutional Animal Care and Use Committee (Approval # 1009009; Date 12/21/2009). Male Sprague–Dawley rats were purchased from Charles River, Wilmington, MA, USA. The animals were adapted to the laboratory environment for 1 week before the experiments. Rats (12 weeks-old) were weighed (264 ± 18 g) and injected intraperitoneally with a single dose of STZ (62 mg/kg; Sigma, St. Louis, MO, USA) dissolved in 0.01 mol/L citric acid solution (pH = 4.5). Control rats received the solvent injected intraperitoneally. The animals were then housed individually in a temperature and humidity-controlled environment on a 12-h light/ dark cycle. To prevent hypoglycemia, the STZ-treated rats could have free access to water with 5% dextrose during the first 24 h after STZ injection. The fasting blood glucose levels in the STZ-treated and control animals were measured 3 days after injection. Those STZ treated animals having a fasting blood glucose level of 300 mg/dL or 11 mmol/L were used. Nicotinamide mononucleotide (NMN; Sigma N3501, St. Louis, MO, USA) was dissolved in sterile saline and injected intraperitoneally into Non-Diabetic and Diabetic rats at a dose of 100 mg/kg on alternate days (Monday, Wednesday, and Friday) at the onset of diabetes for a period of 3 months. Four groups of rats were used: 1. Non-Diabetic; 2. Diabetic; 3. Non-Diabetic + NMN (100 mg/kg); and 4. Diabetic + NMN (100 mg/kg). The total number of rats used for analysis was 24, divided in to 4 groups of 6 rats per group.

### 4.2. Quantification of NAD^+^

A solution of 0.5 N perchloric acid, four times the weight of the hippocampal tissue, was added to the frozen rat hippocampal tissue. These samples were homogenized using a Qiagen tissue lyzer. A 10 μL aliquot of the internal standard solution (50 µg/mL in water) was added to tissue homogenate. The sample was centrifuged at 1600× *g* for 10 min at room temperature. Next, 50 μL of supernatant was transferred to a clean 96-deep-well plate and diluted further with 150 μL of 5 mM ammonium formate, and the NAD^+^ level was quantified following the protocol of Liang et al. [72]. Briefly, NAD^+^ analysis in tissue samples was carried out with a Shimadzu Nexera (MD, USA) coupled to a QTRAP^®^ 5500 mass spectrometer (AB Sciex, Redwood City, CA, USA) equipped with a turbo-electrospray interface in positive ionization mode. The aqueous mobile phase was water with 0.1% formic acid and the organic mobile phase was acetonitrile with 0.1% formic acid. The gradient was 0% formic acid for the first 0.1 min, and then increased to 30% formic acid in 0.9 min, decreased to 0% formic acid within 0.1 min, and maintained at 0% formic acid for another 0.4 min. The flow rate was 0.8 mL/min and the cycle time (injection to injection including instrument delays) was approximately 1.8 min. A volume of 1–3 μL of the final extract was injected onto the analytical dC18 column (100 × 2.1 mm, 3 μm, Milford, Waters, MA, USA).

### 4.3. Magnetic Resonance Imaging (MRI) and Magnetic Resonance Spectroscopy (MRS)

Rats were anesthetized with isoflurane/O_2_ (attached induction and scavenger chamber). Scanning was done on a Bruker Biospec 7.0 Tesla 30 cm horizontal bore scanner using Paravision 5.0 software. A point-resolved spectroscopy (PRESS) pulse sequence (TR/TE = 2500/20 ms) was used for data acquisition from a 3 × 3 × 3 mm^3^ voxel centered around the hippocampus and superior thalamic structures. In the case of MRS spectrum, 300 acquisitions, averaged for a total of 13 min, were completed. At all times during the experiment, the animal was under 1–2% isoflurane anesthesia and oxygen. Three hundred brain MRI acquisitions, averaged for a total of 13 min, were completed in the four groups of rats.

### 4.4. Tissue Processing

At the end of 3 months of NMN treatment, blood glucose level of fasted rats was determined, all the rats were deeply anesthetized with isoflurane and then transcardially perfused with 4% paraformaldehyde in 0.01 M phosphate buffered saline (PBS, pH, 7.4). After perfusion, the brain was removed from the skull. The two hemispheres were separated along the cerebral longitudinal fissure. One of the two hemispheres was sampled randomly, and it was post fixed for 6 h in the same fixative. The brain was processed through a series of sucrose solutions (10% and 20% for 24 h, and 30% for 48 h). Then, the brain was embedded in Optimal Cutting Temperature (OCT) compound for preparing frozen sections.

### 4.5. Estimation of the Volume of Hippocampus, CA1, and DG

Each hemisphere was coronally cut into 1-mm-thick slabs, starting randomly at the rostral pole. A transparent counting grid with an area of 0.36 mm^2^ associated with each point was placed at random on the surface of each slab. The hippocampus volume was calculated according to the Cavalieri’s principle [15,41]. Under the low magnification (10x objective), a counting grid with an area of 1600 µm^2^ was placed at random in each Coomassie blue-stained section. Volumes of CA1 and in DG were counted separately. The stereological workstation consisted of an Olympus microscope, stereology software, and a television screen. The Olympus microscope was equipped with a motorized stage for precise, automatic movements in the X and Y directions. A microcator was attached to the stage for precise measurement of focal depth (in 0.1 µm), and a video camera was used to project images onto the computer screen. The extent of the CA1 layer is defined by the stratum pyramidale on one side and by perforant path fibers on the other [41,42]. The granule cell layer of the DG contains the smallest neuronal cell bodies and is the most densely packed layer in the hippocampus. When sectioned in the transverse plane, the layer appears to have a horseshoe shape. The cell bodies are stacked 8 to 15 cells deep and arranged in radial strands. The total volume of CA1 and DG was calculated following the Cavalieri’s principle [15,41,42].

### 4.6. Estimation of Total Neuronal Numbers in CA1 and DG Regions of the Hippocampus

Serial coronal sections with the thickness of 40 µm were cut using a freezing microtome and mounted onto slides from the rostral end to the caudal end of the rat hemisphere for Coomasie blue staining. For quantification of the number of neurons in the hippocampus, one section was randomly systematically sampled per 15 sections. The number of pyramidal cells and granule cells in CA1 and DG were estimated in the Coomassie blue-stained sections. Under the low magnification (4 × objective) lens, a contour was traced around the edge of the CA1 and DG. Postprocessing section thickness (t) was measured by focusing the top and bottom of the sections (1–3 locations per section). An oil-immersion objective (100 × objective) was used for counting neurons. Under the oil objective, the area of the unbiased counting frame was set as 1600 µm^2^ for neurons in CA1 and in DG. The step length, which was the distance of the dissector moving along the X axis and Y axis from one sampled place to the next one, was set as 400 µm and 400 µm along the X axis and Y axis, respectively, for neurons in CA1 and in DG. The unbiased counting was done as described previously [15,41,42].

### 4.7. Behavioral Memory Tests

We assessed the rats’ memory performance with the Y maze [38]. We performed a spontaneous spatial novelty preference test with the following procedures. The rat started at the end of one arm (start arm), then chose one of the other two arms. One arm of the Y-shaped maze was blocked off, and the rat was able to explore the other two arms for 5 min. The rat was returned to the maze 2 min later with all arms open and kept in the maze for 10 min. The number of entries into each arm was recorded. More entries into the novel arm signify better memory ability. The discrimination ratio of the number of arm entries was calculated with the following formula:(1)$$Discrimination\;ratio\;\%=\left(\frac{novel\;arm}{novel+other\;arm}\right)\times 100$$

In the alternation test, a rat was introduced to the center of the maze. The rat could freely explore the three arms of the Y-shaped maze (A, B, and C) for 15 min. The number of arm entries and the number of triads were recorded to calculate the percentage of alternation. Spontaneous alternation % was then calculated with the following formula: (2)$$Spontaneous\;alternation\;\%=\left(\frac{\#\;spontaneous\;alternations}{total\;number\;of\;arm\;entries-2}\right)\times 100$$

### 4.8. Isolation of Mitochondria

Hippocampal mitochondria from rat brains were isolated using a Percoll (Amersham Biosciences, Piscataway, NJ, USA) gradient centrifugation as described previously [52,53,73,74]. Briefly, hippocampal tissue of a rat’s brain was homogenized in ice-cold mannitol–sucrose (MS) buffer (225 mmol/L mannitol, 75 mmol/L sucrose, 5 mmol/L HEPES, 1 mmol/L EGTA, 1 mg/mL fatty-acid-free bovine serum albumin [BSA], pH 7.4 at 4 °C). The homogenate was centrifuged at 1300× *g* for 3 min, and the pellet was resuspended and centrifuged again at 1300× *g* for 3 min. The pooled supernatants were centrifuged at 22,200× *g* for 8 min, the crude mitochondrial pellet was resuspended in 15% Percoll and then layered on a pre-formed gradient of 40% and 24% Percoll. After centrifugation at 31,700× *g* for 8 min, the mitochondria were collected from the interface of the lower two layers, diluted with isolation medium, and centrifuged at 16,700× *g* for 10 min. The mitochondrial pellet was then resuspended to a final volume of 1.5 mL of MS buffer plus BSA and centrifuged at 22,000× *g* for 10 min at 4 °C in a microfuge. The resulting mitochondrial pellet was resuspended in MS without EGTA. Protein concentrations were determined in triplicate by standard Bradford method employing a BSA standard curve.

### 4.9. Mitochondrial Respiration

Isolated mitochondria were added at a final concentration of 0.5 mg protein/mL to a thermostatically controlled O_2_ electrode chamber (Hansatech Instruments, Norfolk, England) equipped with magnetic stirring and containing 0.5 mL of respiration buffer (125 mmol/L KCl, 20 mmol/L Hepes and 2 mmol/L K2HPO4 at pH 7.4, 37 °C) [38,52,73,74]. Malate and glutamate were used as substrates to assess Complex I-mediated respiration. State 3 respiration was initiated by the addition of 1.0 mmol/L ADP. Approximately 2 min later, State 3 respiration was terminated and State 4 respiration (resting) was initiated with addition of 1.25 µg/mL oligomycin, an inhibitor of the mitochondrial ATP synthase. The maximal rate of uncoupled respiration was subsequently measured by the addition of 65 nmol/L carbonyl cyanide p-(trifluoromethoxy)phenylhydrazone (FCCP), which is a protonophore uncoupling molecule. Rates are an average of 4–6 independent experiments for Complex I respiration.

### 4.10. In-Gel Extraction and Mass Spectroscopy Analysis of Major Acetylated Protein

In-gel extraction of the major acetylated protein was completed as described except hippocampal extract from Diabetic and Non-Diabetic brains was used [45]. Peptides were separated using a 2 h chromatographic gradient online with a data-dependent mass spectroscopy (MS)/MS duty cycle of the top 10 most abundant ions. Database search, peptide quantification, and identification of acetylated lysine-containing peptides were performed using MaxQuant version 1.6.1.0. The excel data of the identified peptide band is described in Appendix A.

### 4.11. Western Blot Analysis

The bilateral hippocampi were quickly extracted from the brain and frozen in liquid nitrogen. The right hippocampi were used for Western blot (6 rats were randomly selected from each group). Hippocampal proteins were extracted with an ice-cold extraction buffer with protease and phosphatase inhibitors. Protein concentration was determined with a BCA kit (BCA Protein Assay Kit, P0010). Protein samples (30 µg) were electrophoresed on SDS-PAGE gels and transferred to PVDF membranes [38,45]. The membranes were blocked with 5% BSA in a TBST buffer and incubated overnight at 4 °C with different primary antibodies: The source and dilution of the various antibodies used in this study were rabbit polyclonal anti-SIRT1 (Millipore 07-131, 1:1000, Burlington, MA, USA), rabbit polyclonal acetylated lysine antibody (Cell Signaling Technology, #9441, 1:1000, Danvers, MA, USA), rabbit polyclonal PGC-1α antibody (Novus Biologicals, #NBP1-04676, 1:1000, Littleton, CO, USA), mouse mAb anti b-actin (Cell Signaling Technology, #3700, 1:1000 Danvers, MA, USA), and rabbit mAb anti GAPDH (Cell Signaling Technology, #5174, 1:1000, Danvers, MA, USA). The intensity was normalized to β-actin. After rinsing with TBST, the membranes were incubated with a secondary antibody, peroxidase-conjugated goat anti-rabbit IgG (H+L). The membranes were developed with an advanced reagent (Bio-Rad, Hercules, CA, USA), and the protein bands were visualized with an automatic chemiluminescence apparatus (Bio-Rad, Hercules, CA, USA). The densities of the bands were determined with the Bio-Rad software.

### 4.12. Statistical Analysis

Comparison of dependent variables was performed on transformed data using factorial ANOVA with a post hoc Tukey test to determine the significance among the groups. Individual comparisons were made using Student’s t-test, assuming unequal variances as previously described [75,76]. Rates between Non-Diabetic and Diabetic hippocampi were statistically compared using a one-way ANOVA test. *p* < 0.05 was considered significant.

## Figures and Tables

**Figure 1 ijms-21-03756-f001:**
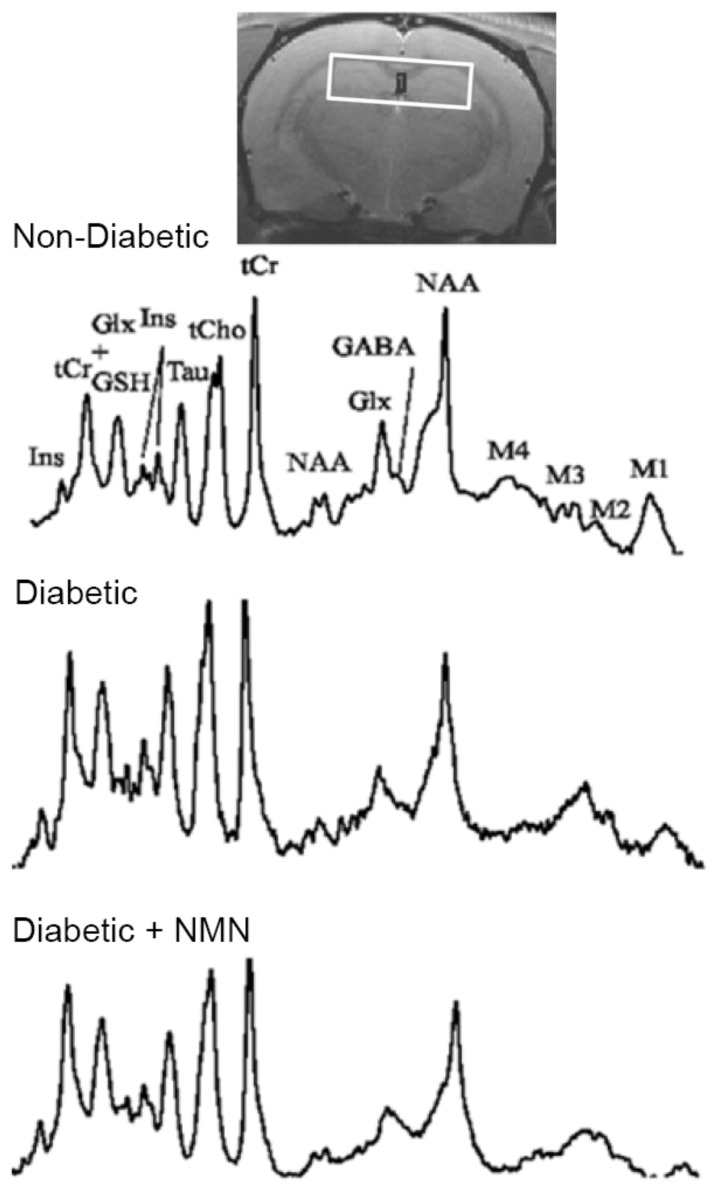
Example of the magnetic resonance spectroscopy spectra acquired at 3 months from Non-Diabetic, Diabetic, Diabetic + NMN, and Non-Diabetic + NMN rats. Only the biochemicals measurable spectra (15%) are reported. GABA = gamma amino butyric acid; Glu = glutamate; Gln = glutamine; PCh = phosphatidyl choline; NAA = N-acetyl aspartate; GSH = glutathione; NAAG = N-acetyl aspartyl glutamate. The data acquired are reported in Table 2.

**Figure 2 ijms-21-03756-f002:**
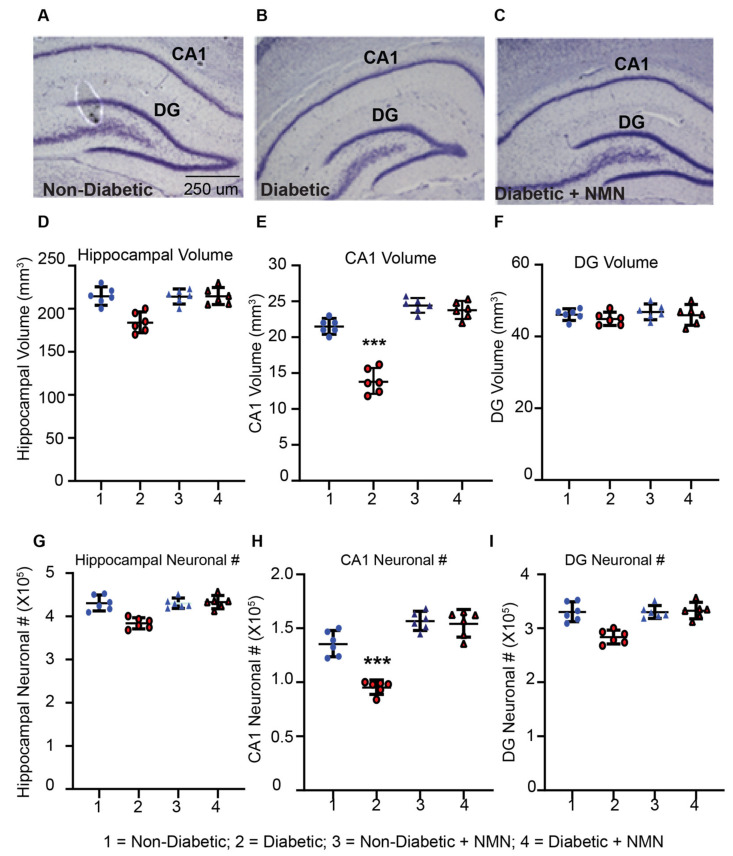
Diabetes-induced loss of hippocampal volume and neuronal counts in CA1 after 3 months of diabetes is prevented by NMN-treatment. (**A**–**C**) Coomassie-blue stained hippocampal section of Non-Diabetic, Diabetic, and Diabetic + NMN rat brains are shown. There was no difference in staining pattern between Non-Diabetic and Non-Diabetic + NMN rats. (**D**) Comparison of the total hippocampal volume between Non-Diabetic and Diabetic rats showed no significant difference. (**E**) Comparison of the CA1 hippocampal neuronal volume between Non-Diabetic and Diabetic rats showed a significant difference (*p* < 0.05) using a one-way ANOVA test with a post hoc Tukey test but no significant difference between hippocampal neuronal volume in Non-Diabetic and Diabetic + NMN rats. (**F**) Comparison of the dentate gyrus (DG) volume between Non-Diabetic and Diabetic rats showed no significant difference between the groups. (**G**) Comparison of the total hippocampal neuronal number (#) between Non-Diabetic and Diabetic rats showed no significant difference. (**H**) Comparison of the CA1 hippocampal neuronal number (#) between Non-Diabetic and Diabetic rats showed a significant difference (*p* < 0.05) using one-way ANOVA test with a post hoc Tukey test but no significant difference between Non-Diabetic and Diabetic + NMN rats. (**I**) Comparison of the DG neuronal number (#) between Non-Diabetic and Diabetic rats showed no significant difference among the groups. *** *p* < 0.001 Diabetes vs. Diabetes + NMN.

**Figure 3 ijms-21-03756-f003:**
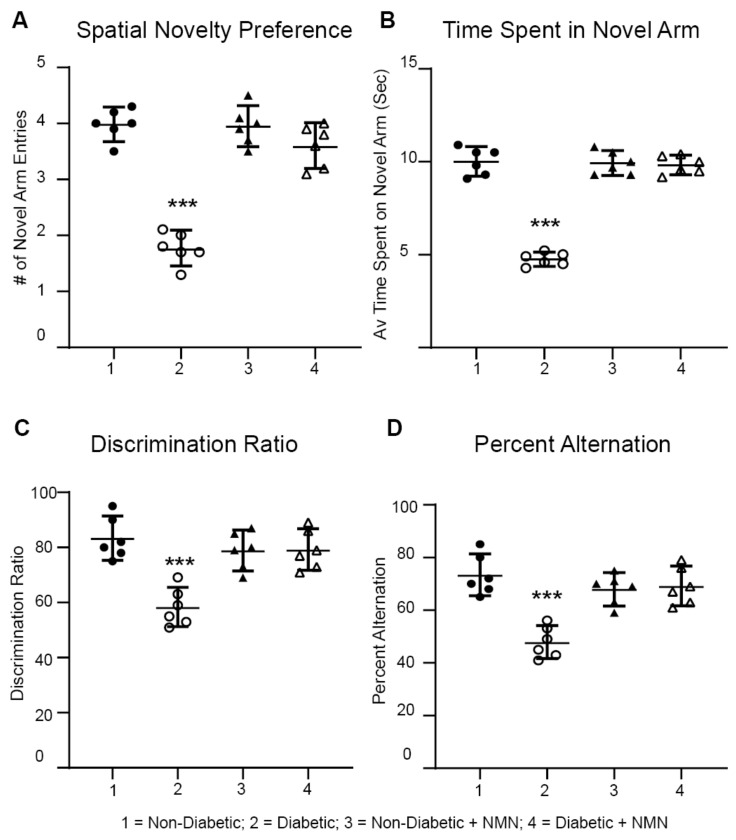
Impaired memory in Diabetic rats was prevented by NMN treatment. Diabetic rats displayed a reduced spontaneous alternation and novel spatial preference compared to those of Non-Diabetic rats at 3 months. (**A**–**C**) Spontaneous spatial novelty preference test: NMN prevents a diabetes-induced decrease in the number (#) of novel arm entries (**A**), the average time spent in the novel arm (**B**), and the discrimination ratio (**C**). (**D**) Spontaneous alternation test: NMN prevents a diabetes-induced decrease in the spontaneous alternation %. (N = 6). *** *p* < 0.001.

**Figure 4 ijms-21-03756-f004:**
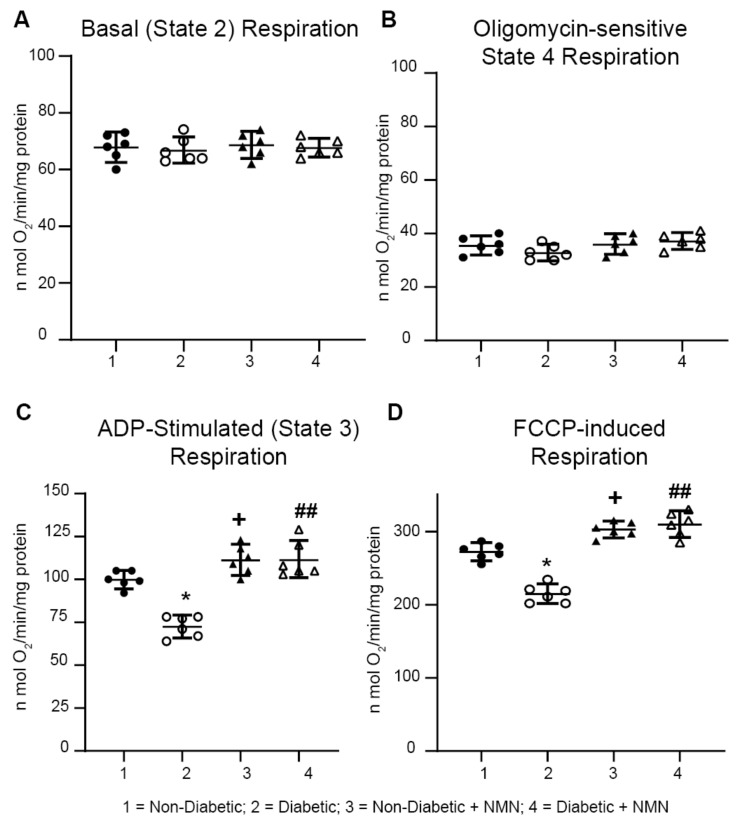
Impaired mitochondrial respiration in Diabetic rats was prevented by NMN treatment. Oxygen consumption rate (OCR) was measured at basal level with the subsequent and sequential addition of ADP, oligomycin, FCCP, and rotenone + antimycin A to hippocampal mitochondria. State 2, State 3, State 4 (oligomycin-sensitive), and FCCP-induced respiration rates were measured. ADP-stimulated and uncoupled respiration was significantly decreased in Diabetic rats compared to Non-Diabetic rats. Administration of NMN significantly increased both ADP-stimulated and uncoupled respiration. Spare respiratory capacity was calculated after subtracting the basal respiration from uncoupled respiration and was significantly higher (*p* < 0.05) in NMN treated Non-Diabetic and Diabetic hippocampal mitochondria. * *p* < 0.05 Diabetic vs. Non-Diabetic; + *p* < 0.05 Non-Diabetic vs. Non-Diabetic + NMN; ## *p* < 0.01 Diabetic vs. Diabetic + NMN.

**Figure 5 ijms-21-03756-f005:**
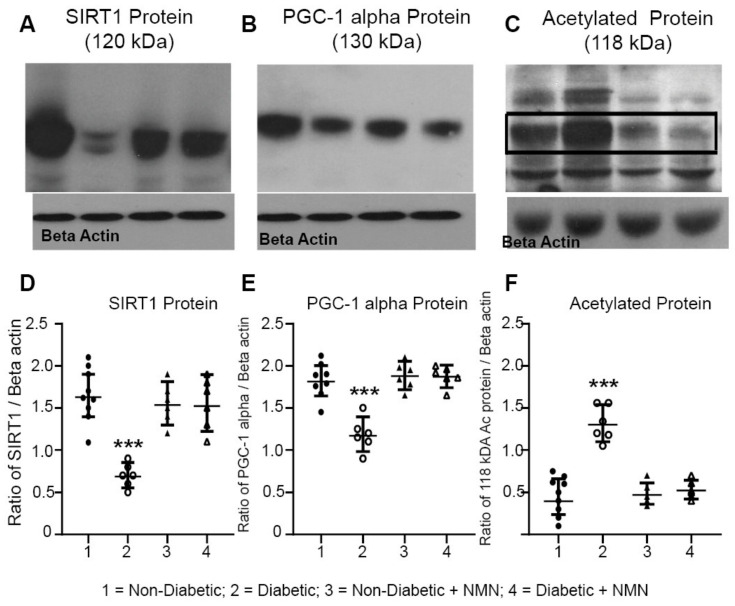
(**A**–**C**) Western blot analysis of NAD^+^-consuming enzyme SIRT1 and acetylated proteins. Hippocampal protein extracts were prepared from Non-Diabetic, Diabetic, Non-Diabetic + NMN, and Diabetic + NMN rat brains. Preparation of hippocampal protein extracts, blot analysis, the source, and the dilution of the antibodies used are described in Section 4. (**D**–**F**) Quantification of the intensity of the bands are shown. Significant decrease in SIRT1 protein and PGC-1α protein levels, and an increase in acetylated 118 kDa protein were observed in Diabetic samples, but not in NMN-treated Diabetic + NMN samples. *** *p* < 0.001 Diabetic vs. Diabetic + NMN.

**Table 1 ijms-21-03756-t001:** Metabolic end points and brain nicotinamide adenine dinucleotide (NAD^+^) levels in Diabetic (Dia) and Non-Diabetic (Non-Dia) rats + nicotinamide mononucleotide (NMN) (100 mg/day).

Metabolite	Non-Dia (*n* = 6) 1	Dia (*n* = 6) 2	Non-Dia + NMN (*n* = 6) 3	Dia + NMN (*n* = 6) 4	1 vs. 2	2 vs. 4	1 vs. 3	3 vs. 4	1 vs. 4
**Body Weight (g)**	416 ± 26	318 ± 21	421 ± 32	296 ± 14	<0.05	NS	NS	<0.05	<0.05
**Glucose (mM)**	6.1 ± 0.6	22.7 ± 2	6 ± 0.4	24.6 ± 1.3	<0.001	NS	NS	<0.001	<0.001
**Cholesterol (mg/dL)**	172 ± 14	166 ± 14	165 ± 12	163.4 ± 22	NS	NS	NS	NS	NS
**Triglycerides (mg/dL)**	185 ± 6	120 ± 3	177 ± 8	124 ± 6	<0.001	NS	NS	<0.001	<0.001
**GTT-AUC (mg/dL/min)**	10472 ± 200	30030 ± 300	9989 ± 190	29854 ± 278	<0.001	NS	NS	<0.001	<0.001
**Brain NAD^+^ (pmol/mg)**	200 ± 15	123 ± 15	237 ± 13	198 ± 12	<0.05	<0.05	NS	<0.05	NS

Data represent the mean value ± SEM. GTT-AUC = glucose tolerance test-area under the curve; NS = not significant.

**Table 2 ijms-21-03756-t002:** MRS hippocampal metabolites in Diabetic (Dia) and Non-Diabetic (Non-Dia) Rats + NMN (100 mg/day).

Metabolite mM	Non-Dia 1	Dia 2	Non-Dia + NMN 3	Dia + NMN 4	1 vs. 2	2 vs. 4	1 vs. 3	3 vs. 4	1 vs. 4
**GABA**	0.29	0.33	0.27	0.27	0.018	0.01	NS	NS	NS
**Glucose**	0.61	1.07	0.60	0.97	<0.001	NS	NS	0.002	0.002
**Glutamine**	0.38	0.37	0.36	0.36	NS	NS	NS	NS	NS
**Glutamate**	1.16	1.31	1.14	1.14	<0.001	<0.001	NS	NS	NS
**Ph. Choline**	0.14	0.12	0.14	0.15	NS	NS	NS	NS	NS
**Myoinositol**	1.00	1.15	1.00	1.16	0.018	NS	NS	NS	0.014
**NAA**	1.22	1.03	1.23	1.19	0.002	0.003	NS	NS	NS
**Taurine**	0.81	0.91	0.79	0.87	<0.001	0.010	NS	0.04	0.007
**GSH**	0.25	0.17	0.27	0.21	<0.001	0.006	NS	0.03	0.009
**GPC + PCh**	0.18	0.17	0.17	0.19	NS	NS	NS	NS	NS
**NAA + NAAG**	1.17	1.19	1.14	1.14	NS	0.013	NS	NS	NS
**Glu + Gln**	1.54	1.67	1.51	1.50	0.031	0.008	NS	NS	NS

MRS = magnetic resonance spectroscopy. Values are expressed as a mean ratio to creatine + phosphocreatine; Glu = glutamate; Gln = glutamine; PCh = phosphatidyl choline; NAA = N-acetyl aspartate; GSH = glutathione; NAAG = N-acetyl aspartyl glutamate. NS = not significant.

**Table 3 ijms-21-03756-t003:** Hippocampal mitochondrial function in Diabetic (Dia) and Non-Diabetic (Non-Dia) Rats + NMN (100 mg/day).

nmol O_2_ /min/mg Protein	Non-Dia 1	Dia 2	Non-Dia + NMN 3	Dia + NMN 4	1 vs. 2	2 vs. 4	1 vs. 3	3 vs. 4	1 vs. 4
**State 2 Respiration**	63.8 ± 4.8	68.6 ± 4.3	66.8 ± 4.3	69.6 ± 2.9	NS	NS	NS	NS	NS
**State 3 Respiration**	101.8 ± 5.3	82.2 ± 11.8	119.8 ± 7.5	109.3 ± 5.6	<0.05	<0.01	<0.05	NS	NS
**State 4 Respiration**	35.5 ± 3.3	36 ± 3.5	32.8 ± 2.8	37.2 ± 2.9	NS	NS	NS	NS	NS
**FCCP-Induced Respiration**	272 ± 11.3	240 ± 13.6	302.7 ± 10.3	295 ± 16.8	<0.05	<0.01	<0.05	NS	<0.05
**Spare Reserve Capacity**	204 ± 15.1	162 ± 16	235 ± 11	242 ± 17.7	<0.05	<0.01	<0.05	NS	<0.05

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
