# Peer review of "Nicotinamide Mononucleotide Administration Prevents Experimental Diabetes-Induced Cognitive Impairment and Loss of Hippocampal Neurons"

_ijms, 2020, doi:10.3390/ijms21113756_

Round 1
Reviewer 1 Report
The Authors in study no ijms-794793 examined if the administration of the precursor of NAD +, nicotinamide mononucleotide (NMN), can prevent diabetes rats memory deficits.
As reported by the World Health Organization (WHO) currently there are 387 million people worldwide suffering from diabetes, which makes 8.3% of the global human population. Along with the increase in the prevalence of diabetes the number of patients with complications will rise significantly. Untreated or poorly controlled diabetes can increase the risk of serious complications, the therapy of which is a great burden to healthcare systems all over the world. Therefore, the authors' research is justified. The studies presented to me for review were well planned out. The authors have used numerous research methods, among others, the volume of the hippocam was estimated and the total number of neurons in the hippocampus was examined histopathologically. Western Blot, Magnetic Resonance Imaging (MRI) as well as Behavioral Memory Tests were used.
There are a few small mistakes which should be corrected:
- The number of animals tested is not clear. On page 3, line 97, the authors wrote:
There were 4 groups (n = 6): 1: Non-Diabetic; 2: Diabetic; 3: Non-Diabetic + NMN and 4: Diabetic + NMN.
In Table 1, each column has n = 6 x 4 groups, which gives 24.
2. Lack of information on the number of animals tested also on page 14, Materials and Methods , 4.1. Animal Procedure –
Four groups of rats were used; 1. Non-Diabetic; 2. Diabetic; 3. Non-Diabetic + NMN (100 mg/kg); and 4. Diabetic + 418 NMN (100 mg/kg)
- There is not information about the procedure the authors used a dose of nicotinamide mononucleotide (100 mg/ kg) rat weight. Please specify
- I suggest that authors use the expression: Diabetic rats / Non-Diabetic rats instead of Diabetic group / Non-Diabetic group throughout the manuscript e.g
lines 309 -314
Measurement of the effects of diabetes and administration of NMN on the metabolic profile showed that levels of hippocampal biochemicals GABA/total Cr, Glutamate/total Cr, Glutamate + Glutamine/total Cr, NAA/total Cr, Taurine/total Cr and mI/total Cr were significantly altered in the Diabetic group compared to the Non-Diabetic group, suggesting that energy metabolism, neurotransmission, and lipid membrane metabolism were impaired in the Diabetic group [47,63,64].
lines 325-327
Accordingly, our unbiased stereological analysis and behavioral measurement showed a significant cell loss in the CA1 region and cognitive impairment in the Diabetic compared to the Non-Diabetic group. etc.
- The manuscript is "overloaded" with the number of citations (a total of 106 items), which are not always well founded.
For example
Our results confirm the previous studies that showed the same changes in biochemicals in the brains of Type 1 and Type 2 models of diabetes [14,46-48]. The metabolic profile showed that administration of NMN to diabetic rats decreased glutamate, myoinositol, taurine and NAA; increased GSH and normalized the diabetes-induced changes in the biochemicals except that NMN did not affect hipocampal glucose levels and remained at the diabetic level [Diabetic vs. Diabetic + NMN (1.07 vs. 0.97; NS); Table 2].
Citations (46-48) relate to patients with type 1 or type 2 diabetes, not animal testing. Please change it.
Author Response
(x) I would not like to sign my review report
( ) I would like to sign my review report
English language and style
( ) Extensive editing of English language and style required
( ) Moderate English changes required
( ) English language and style are fine/minor spell check required
(x) I don't feel qualified to judge about the English language and style
Yes |
Can be improved |
Must be improved |
Not applicable |
|
Does the introduction provide sufficient background and include all relevant references? |
(x) |
( ) |
( ) |
( ) |
Is the research design appropriate? |
(x) |
( ) |
( ) |
( ) |
Are the methods adequately described? |
( ) |
(x) |
( ) |
( ) |
Are the results clearly presented? |
( ) |
( ) |
(x) |
( ) |
Are the conclusions supported by the results? |
( ) |
(x) |
( ) |
( ) |
Comments and Suggestions for Authors
The Authors in study no ijms-794793 examined if the administration of the precursor of NAD +, nicotinamide mononucleotide (NMN), can prevent diabetes rats memory deficits.
As reported by the World Health Organization (WHO) currently there are 387 million people worldwide suffering from diabetes, which makes 8.3% of the global human population. Along with the increase in the prevalence of diabetes the number of patients with complications will rise significantly. Untreated or poorly controlled diabetes can increase the risk of serious complications, the therapy of which is a great burden to healthcare systems all over the world. Therefore, the authors' research is justified. The studies presented to me for review were well planned out. The authors have used numerous research methods, among others, the volume of the hippocam was estimated and the total number of neurons in the hippocampus was examined histopathologically. Western Blot, Magnetic Resonance Imaging (MRI) as well as Behavioral Memory Tests were used.
There are a few small mistakes which should be corrected:
- The number of animals tested is not clear. On page 3, line 97, the authors wrote:
There were 4 groups (n = 6): 1: Non-Diabetic; 2: Diabetic; 3: Non-Diabetic + NMN and 4: Diabetic + NMN.
In Table 1, each column has n = 6 x 4 groups, which gives 24.
Response: Corrected the wording to say (n=6 for each group, total = 24):
- Lack of information on the number of animals tested also on page 14, Materials and Methods , 4.1. Animal Procedure –
Four groups of rats were used; 1. Non-Diabetic; 2. Diabetic; 3. Non-Diabetic + NMN (100 mg/kg); and 4. Diabetic + 418 NMN (100 mg/kg)
Response: Corrected the wording to say: The total number of rats used for analysis was 24, divided in to 4 groups of 6 rats per group.
- There is not information about the procedure the authors used a dose of nicotinamide mononucleotide (100 mg/ kg) rat weight. Please specify
Response: Nicotinamide mononucleotide (NMN; Sigma N3501) was dissolved in sterile saline and injected intraperitoneally into non-diabetic and diabetic rats at a dose of 100 mg/kg on alternate days (Monday, Wednesday and Friday) for a period of 3 months [1].
Lines 415 to 418.
- I suggest that authors use the expression: Diabetic rats / Non-Diabetic rats instead of Diabetic group / Non-Diabetic group throughout the manuscript e.g lines 309 -314
Response: We changed non-diabetic and diabetic groups to non-diabetic and diabetic rats, lines 309 to 357.
Measurement of the effects of diabetes and administration of NMN on the metabolic profile showed that levels of hippocampal biochemicals GABA/total Cr, Glutamate/total Cr, Glutamate + Glutamine/total Cr, NAA/total Cr, Taurine/total Cr and mI/total Cr were significantly altered in the Diabetic group compared to the Non-Diabetic group, suggesting that energy metabolism, neurotransmission, and lipid membrane metabolism were impaired in the Diabetic group [47,63,64].
lines 325-327
Response: We changed non-diabetic and diabetic groups to non-diabetic and diabetic rats, in the entire manuscript
Accordingly, our unbiased stereological analysis and behavioral measurement showed a significant cell loss in the CA1 region and cognitive impairment in the Diabetic compared to the Non-Diabetic group. etc.
Response: We changed non-diabetic and diabetic groups to non-diabetic and diabetic rats, in the entire manuscript.
- The manuscript is "overloaded" with the number of citations (a total of 106 items), which are not always well founded.
For example
Our results confirm the previous studies that showed the same changes in biochemicals in the brains of Type 1 and Type 2 models of diabetes [14,46-48]. The metabolic profile showed that administration of NMN to diabetic rats decreased glutamate, myoinositol, taurine and NAA; increased GSH and normalized the diabetes-induced changes in the biochemicals except that NMN did not affect hipocampal glucose levels and remained at the diabetic level [Diabetic vs. Diabetic + NMN (1.07 vs. 0.97; NS); Table 2].
Citations (46-48) relate to patients with type 1 or type 2 diabetes, not animal testing. Please change it.
Response: We tightened the references to cite well founded references.
Submission Date
21 April 2020
Date of this review
Reviewer 2 Report
In the present manuscript, authors intended to evaluate the effectiveness of nicotinamide mononucleotide (NMN) in reverting and/or ameliorating diabetes-mediated brain effects, in particular those ocurring within hippocampus. Even though the manuscript presents interesting results and the tecniques used are appropriate, there are some inconsistencies throughout the manuscript that deserve further attention.
1 - How was NMN administered to rats? It is not clear because in different parts of the manuscript we can read different information. For instance, line 396: "In summary our results show that subcutaneous administration of NMN..." and in line 416 "Nicotinamide mononucleotide (NMN) was dissolved in saline and injected intraperitoneally into diabetic rats..."
2 - Why did authors chose the concentration of 100 mg/kg of NMN?
3 - Regarding western blot, did authors used the same membrane to detect the three different antibodies? Looking at the representative images of actin, it is visible that the membrane is always the same. Please check this.
4 - Regarding Fig. 4, please change the designation of "Basal (state 4) Respiration" to State 2. As widely described, state 4 of respiration refers to the respiration state of mitochondria after the addition of oligomycin and not the basal respiration of mitochondria.
5 - Also in the same figure, the order should be changed. Fig.4A should be the State 2; Fig.4B the State 3; Fig.4C the State 4; and Fig.4D the FCCP-induced respiration.
6 - In this figure and in table 1 is visible that NMN promoted a significant increase in mitochondrial state 3, uncoupled respiration and spare reserve capacity in both non-diabetic and diabetic rats. Do authors have an explanation to this situation?
7 - In discussion, section 3.5, authors concluded that "We interpret these results to suggest that administration of NMN primes the mitochondria by increasing their reserve capacity to combat mitochondrial stress under diabetic conditions". However, considering that this also ocurred in the non-diabetic rats, that cannot be the appropriate conclusion.
8 - In abstract, we can read that NMN increased the brain levels of NAD+, however looking at table 1, NAD+ levels remain statistically unchanged when comparing diabetic and NMN-treated diabetic rats. Please check this.
9 - In the same table, in some places we have <0.001 and in the line of NAD+ appears the real value. Please normalize.
10 - In results section 2.6, we can read " One of the major de-acetylated protein bands (~102 kDa) was cut and subjected to mass spectrometry
analysis." Where are the results of this experiment?
11 - In discussion section, we can read "We propose that improved mitochondrial respiratory capacity via PGC-1α and associated mitophagy via NEDD4 act synergistically to provide neuroprotection. " There are no results in the manuscript regarding mitophagy that allow this suggestion. Please check this.
12 - In discussion section 3.3, we can read "Subcutaneous administration of NMN (100mg/day) on alternate days to diabetic rats normalized a diabetes-induced decrease in brain NAD+ levels in rats (Group 3 versus Group 4 in Table 1)". However, in table 1 and in all the figures, group 3 refers to the non-diabetic treated rats and group 4 refers to diabetic-treated rats.
13 - Considering that most of the results obtained with NMN treatment ocurred in both diabetic and non-diabetic rats, it seems that the conclusion should be rearrenged and cannot be only focused in the diabetic group.
Author Response
29 Apr 2020 12:39:43
Open Review
(x) I would not like to sign my review report
( ) I would like to sign my review report
English language and style
( ) Extensive editing of English language and style required
( ) Moderate English changes required
(x) English language and style are fine/minor spell check required
( ) I don't feel qualified to judge about the English language and style
Yes |
Can be improved |
Must be improved |
Not applicable |
|
Does the introduction provide sufficient background and include all relevant references? |
(x) |
( ) |
( ) |
( ) |
Is the research design appropriate? |
( ) |
(x) |
( ) |
( ) |
Are the methods adequately described? |
(x) |
( ) |
( ) |
( ) |
Are the results clearly presented? |
( ) |
(x) |
( ) |
( ) |
Are the conclusions supported by the results? |
( ) |
( ) |
(x) |
( ) |
Comments and Suggestions for Authors
In the present manuscript, authors intended to evaluate the effectiveness of nicotinamide mononucleotide (NMN) in reverting and/or ameliorating diabetes-mediated brain effects, in particular those ocurring within hippocampus. Even though the manuscript presents interesting results and the tecniques used are appropriate, there are some inconsistencies throughout the manuscript that deserve further attention.
1 - How was NMN administered to rats? It is not clear because in different parts of the manuscript we can read different information. For instance, line 396: "In summary our results show that subcutaneous administration of NMN..." and in line 416 "Nicotinamide mononucleotide (NMN) was dissolved in saline and injected intraperitoneally into diabetic rats..."
Response: NMN was administered in rats by intraperitoneal injection. We had used subcutaneous injection of NMN in mice, hence the confusion, corrected in the revised manuscript.
2 - Why did authors chose the concentration of 100 mg/kg of NMN?
Response: In earlier studies NMN was used by injecting intraperitoneally at a concentration of 500 mg/kg to demonstrate its protective effect in diabetic/ischemic/Alzheimer’s disease mouse models (1 - 3). Later studies used intraperitoneal injection at a concentration of 100 mg/kg/ and showed protective effects on brain mitochondrial function (4,5). In a pilot study, we tested 2 concentrations of NMN (100 mg/kg and 250 mg/kg) and found similar increase in brain NAD+ levels. Therefore, we used intraperitoneal injection of 100 mg/kg in all our experiments. Now NMN is used as a dietary supplement (6). In a new study Grozio et al. Identified a previously characterized amino acid and polyamine transporter called Slc12a8 as an NMN transporter in the gut endothelium (7). Clinical study showed that NMN (oral dose 500 mg) was safe and effectively metabolized in healthy men without causing any significant deleterious effects (8).
- Yoshino, J.; Mills, K.F.; Yoon, M.J.; Imai, S. Nicotinamide mononucleotide, a key NAD(+) intermediate, treats the pathophysiology of diet- and age-induced diabetes in mice. Cell Metab. 2011; 14, 528–536.
NMN dose: a bolus intraperitoneal administration of NMN (500 mg/kg body weight) increased tissue NMN and NAD+ levels.
- Yamamoto, T.; Byun, J.; Zhai, P.; Ikeda, Y.; Oka, S.; Sadoshima, J. Nicotinamide Mononucleotide, an Intermediate of NAD+ Synthesis, Protects the Heart from Ischemia and Reperfusion. PLoS ONE. 2014;9, e98972.
NMN dose 500 mg/kg injection i.p.
- Wang X, Hu X, Yang Y, Takata T, Sakurai T. Brain Res. 2016;1643, 1-9.
NMN dose 500 mg/kg injection i.p.
- Long, A.N.; Owens, K.; Schlappal, A.E.; Kristian, T.; Fishman, P.S.; Schuh, R.A. Effect of nicotinamide mononucleotide on brain mitochondrial respiratory deficits in an Alzheimer’s disease-relevant murine model. BMC Neurol. 2015; 15, 19.
NMN dose 100 mg/kg injection i.p.
- Hosseini L, Vafaee MS, Badalzadeh R. J Cardiovasc Pharmacol Ther, 2020;25(3):240-250.
NMN dose 100 mg/kg injection i.p.
- Mills, K.F.; Yoshida, S.; Stein, L.R.; Grozio, A.; Kubota, S.; Sasaki, Y.; Redpath, P.; Migaud, M.E.; Apte, R.S.;Uchida, K.; et al. Long-Term Administration of Nicotinamide Mononucleotide Mitigates Age-Associated Physiological Decline in Mice. Cell Metab. 2016; 24, 795–806.
Oral administration of 100 and 300 mg to mice.
- Grozio A, Mills KF, Yoshino J, Bruzzone S, Sociali G, Tokizane K, Lei HC, Cunningham R, Sasaki Y, Migaud ME, Imai SI. Nat Metab. 2019; 1, 47-57.
- Irie J, Inagaki E, Fujita M, Nakaya H, Mitsuishi M, Yamaguchi S, Yamashita K, Shigaki S, Ono T, Yukioka H, Okano H, Nabeshima YI, Imai SI, Yasui M, Tsubota K, Itoh H. Endocr J. 2020;67(2):153-160.
3 - Regarding western blot, did authors used the same membrane to detect the three different antibodies? Looking at the representative images of actin, it is visible that the membrane is always the same. Please check this.
Response: It was a mistake. We used the same blot to probe SIRT1 and PGC-1α, and a third blot for Acetylated protein. We have modified the figure to reflect this, the quantification as a ratio to beta actin was done accordingly.
4 - Regarding Fig. 4, please change the designation of "Basal (state 4) Respiration" to State 2. As widely described, state 4 of respiration refers to the respiration state of mitochondria after the addition of oligomycin and not the basal respiration of mitochondria.
Response: Changed the designation of "Basal (state 4) Respiration" to State 2.
5 - Also in the same figure, the order should be changed. Fig.4A should be the State 2; Fig.4B the State 3; Fig.4C the State 4; and Fig.4D the FCCP-induced respiration.
Response: Changed the order as suggested: Fig.4A was changed to State 2; Fig.4B Oligomycin-sensitive to State 4; Fig.4C ADP-stimulated to State 3; and Fig.4D to FCCP-induced respiration. Changed it also in the figure legend. Lines 228-230.
6 - In this figure and in table 1 is visible that NMN promoted a significant increase in mitochondrial state 3, uncoupled respiration and spare reserve capacity in both non-diabetic and diabetic rats. Do authors have an explanation to this situation?
Response: Our explanation is: State 3 is controlled exclusively by substrate oxidation and will detect dysfunction in respiratory chain components, substrate translocases or dehydrogenases. The reserve respiratory capacity is the difference between ATP produced by oxidative phosphorylation at basal and at maximal respiratory activity. When the need for cellular energy arises, mitochondria respond to increase ATP, thereby increasing the respiratory rate (State 3). Upon a strong challenge, these mitochondria then reach a near maximal rate to uphold energy requirements (uncoupled maximal respiration). Regulation of oxidative phosphorylation can be divided into short- and long-term. Short-term regulators mediate the oxidative phosphorylation according to the immediate energy requirement. Short-term regulators are, therefore, important for the regulation of the ATP production. Long-term regulators of the oxidative phosphorylation modulate the properties of the oxidative phosphorylation and are correspondingly essential for the setting of the maximal respiratory capacity. The increase in State 3, uncoupled respiration, and spare reserve respiratory capacity by NMN treatment is likely to reflect the interplay between the short- and long-term regulators of the oxidative phosphorylation. Similar results of increased State 3 (~15%) and uncoupled respiration (~20%) has been obtained after NMN treatment (100 mg/kg) in the brains of an Alzheimer’s Disease (AD) mouse model ( [1] lines 373-381.
7 - In discussion, section 3.5, authors concluded that "We interpret these results to suggest that administration of NMN primes the mitochondria by increasing their reserve capacity to combat mitochondrial stress under diabetic conditions". However, considering that this also ocurred in the non-diabetic rats, that cannot be the appropriate conclusion.
Response: The reviewer is right. Revised the sentence “We interpret these results to suggest that administration of NMN primes the mitochondria by increasing their reserve capacity to combat mitochondrial stress under both non-diabetic and diabetic conditions. line 371-373.
8 - In abstract, we can read that NMN increased the brain levels of NAD+, however looking at table 1, NAD+ levels remain statistically unchanged when comparing diabetic and NMN-treated diabetic rats. Please check this.
Response: The reviewer is right, NAD+ levels is statistically significant (P<0.05) when comparing diabetic and NMN-treated diabetic rats
9 - In the same table, in some places we have <0.001 and in the line of NAD+ appears the real value. Please normalize.
Response: Corrected the statistics to represent *P<0.05, ** P<0.01 and *** P<0.001.
10 - In results section 2.6, we can read " One of the major de-acetylated protein bands (~102 kDa) was cut and subjected to mass spectrometry analysis." Where are the results of this experiment?
Response: The experiment is included in the Materials and Methods Section
4.10. In-gel extraction and mass spectroscopy analysis of major acetylated protein
11 - In discussion section, we can read "We propose that improved mitochondrial respiratory capacity via PGC-1α and associated mitophagy via NEDD4 act synergistically to provide neuroprotection. " There are no results in the manuscript regarding mitophagy that allow this suggestion. Please check this.
Response: Revised it SIRT1 regulates mitochondrial function in hippocampal neurons through PGC1-alpha, activators or overexpression of SIRT11 has been shown to regulate axonal growth [2-4]. Line 274-276.
Added the following in the discussion In addition, in mice that express enhanced yellow fluorescent protein (eYFP) in neuronal mitochondria, administration of NMN had far less fragmented mitochondria, but had longer neuronal mitochondria in the CA1 region, suggesting that NMN treatment gives rise to longer mitochondria in the hippocampal sub-region, likely through increased fusion and/or decreased fission [1]. Line 402-405.
12 - In discussion section 3.3, we can read "Subcutaneous administration of NMN (100mg/day) on alternate days to diabetic rats normalized a diabetes-induced decrease in brain NAD+ levels in rats (Group 3 versus Group 4 in Table 1)". However, in table 1 and in all the figures, group 3 refers to the non-diabetic treated rats and group 4 refers to diabetic-treated rats.
Response: Corrected the wording: "Intraperitoneal administration of NMN (100mg/day) on alternate days to diabetic rats normalized a diabetes-induced decrease in brain NAD+ levels in rats (Group 2 versus Group 4 in Table 1)". Lines 300-302.
13 - Considering that most of the results obtained with NMN treatment occurred in both diabetic and non-diabetic rats, it seems that the conclusion should be rearranged and cannot be only focused in the diabetic group.
Response: Conclusion was revised: In summary our results show that intraperitoneal administration of NMN increased the levels of NAD+ in both non-diabetic and diabetic brains. Diabetes, on the other hand decreased brain NAD+ levels and that administration of NMN compensated the loss of brain NAD+ and thereby prevented diabetic cognitive impairment and diabetes-induced loss of hippocampal neurons that paralleled preservation of mitochondrial bioenergetic function and activation of the SIRT1 pathway. The study suggests that medications, which prevent NAD+ depletion induced by diabetes mellitus and molecules that activate SIRT1, may provide a therapeutic intervention for diabetic neurodegeneration in the central nervous system.
Submission Date
21 April 2020
Date of this review
04 May 2020 18:06:08
- Long, A.N.; Owens, K.; Schlappal, A.E.; Kristian, T.; Fishman, P.S.; Schuh, R.A. Effect of nicotinamide mononucleotide on brain mitochondrial respiratory deficits in an Alzheimer's disease-relevant murine model. BMC Neurol 2015, 15, 19, doi:10.1186/s12883-015-0272-x.
- Li, X.H.; Chen, C.; Tu, Y.; Sun, H.T.; Zhao, M.L.; Cheng, S.X.; Qu, Y.; Zhang, S. Sirt1 promotes axonogenesis by deacetylation of Akt and inactivation of GSK3. Mol Neurobiol 2013, 48, 490-499, doi:10.1007/s12035-013-8437-3.
- Chandrasekaran, K.; Salimian, M.; Konduru, S.R.; Choi, J.; Kumar, P.; Long, A.; Klimova, N.; Ho, C.Y.; Kristian, T.; Russell, J.W. Overexpression of Sirtuin 1 protein in neurons prevents and reverses experimental diabetic neuropathy. Brain 2019, 142, 3737-3752, doi:10.1093/brain/awz324.
- Roy Chowdhury, S.K.; Smith, D.R.; Saleh, A.; Schapansky, J.; Marquez, A.; Gomes, S.; Akude, E.; Morrow, D.; Calcutt, N.A.; Fernyhough, P. Impaired adenosine monophosphate-activated protein kinase signalling in dorsal root ganglia neurons is linked to mitochondrial dysfunction and peripheral neuropathy in diabetes. Brain 2012, 135, 1751-1766, doi:10.1093/brain/aws097.
Round 2
Reviewer 2 Report
In general authors tried to provide proper answers to the majority of reviewers suggestions. However, there are some answers/alterations that were not performed in the manuscript.
1 - Authors answered "We used the same blot to probe SIRT1 and PGC-1α, and a third blot for Acetylated protein. We have modified the figure to reflect this, the quantification as a ratio to beta actin was done accordingly." However, Fig.5 remains unchanged without the mentioned alterations.
2 - Also, authors answered "NAD+ levels is statistically significant (P<0.05) when comparing diabetic and NMN-treated diabetic rats". However, looking at table 1, the comparison between group 2 (diabetic) vs group 4 (NMN-treated diabetic rats) gives a NS significance. There seems to be a confusion regarding this. Please check.
3 - Where are the results of this technique "4.10. In-gel extraction and mass spectroscopy analysis of major acetylated protein"?
4 - Figure 4A was not changed as supposed. There is still the designation Basal (State 4).
5 - Also, in the same figure, Fig.4B should be State 3 and Fig.4C should be State 4.
